# Trends in stroke admissions before, during and post-peak of the COVID-19 pandemic: A one-year experience from the Qatar stroke database

Naveed Akhtar[1], Saadat Kamran[1], Salman Al-Jerdi[2], Yahia Imam[1], Sujatha Joseph[1], Deborah Morgan[1], Mohamed Abokersh[1], R. T. Uy[1], Ashfaq Shuaib[3]*

1 The Neuroscience Institute, Hamad Medical Corporation, Doha, Qatar, 2 Weill Cornell Medical College-Qatar Foundation, Doha, Qatar, 3 Neurology Division, Department of Medicine, University of Alberta, Edmonton, Canada

* ashfaq.shuaib@ualberta.ca

## Abstract

### Background

Several reports document a decrease in the rates of stroke hospital admissions during the covid-19 pandemic. There is very little information whether the admission rates will change as the infection is controlled. We report on our rates of admissions before, during and following the peak of covid-19 infections in a prospective database from Qatar.

### Methods and results

The stroke admissions in the six months prior to COVID-19 pandemic averaged 229/month. There was a decrease to 157/month in March-June during the peak of the pandemic. In the 6 months following the peak, as covid-19 numbers began to decrease, the average numbers increased back to 192/month. There was an increase in severe ischemic strokes and decreased in functional recovery. The decreased admissions were mainly driven by fewer stroke mimics. Patients presenting with ischemic stroke or cerebral hemorrhage remained unchanged.

### Conclusions

Fewer stroke mimics presenting to the hospital can explain the fewer admissions and poor outcome at the height of the covid-19 pandemic. The continued decrease in the number of ischemic stroke and stroke mimic admissions following the pandemic peak requires more study.

**Data Availability Statement:** All relevant data are in the paper and supporting information files.

**Funding:** Dr.Naveed Akhtar received funding for the grant 16016/16 from Hamad Medical Corporation. No role of the sponsor in the study.

**Competing interests:** The authors have declared that no competing interests exist.

## Introduction

There are a number of reports from around the world have documented a decrease in the number of stroke related admission during the peak months of the first wave of the COVID-19 pandemic [1–15]. The decrease has been most apparent with transient ischemic attacks, minor stroke and stroke mimics [4, 8, 11, 13, 14]. We have recently shown a significant decrease in stroke admissions to the Hamad General Hospital (HGH) in Qatar during the COVID-19 pandemic that was predominantly driven by a decrease in transient ischemic attacks (TIAs), minor stroke and stroke mimics [14]. The decrease has however not been universally reported, with some centers reporting no decrease in admission rates [16]. A reluctance to seek medical attention for fear of contracting the virus may be contributing to the decreased frequency in such categories. Strict instructions to stay at home during lockdowns may contribute to this trend. Social isolation, especially in the elderly may allow for a lack of detection by family members who are not able to make the diagnosis. Secondly, the emergency services may be overwhelmed by responses to the pandemic, thus neglecting the stroke patients [17]. The risk of recurrent stroke is highest following a TIA and there is considerable evidence that this is reduced significantly with appropriate assessment and treatment [18]. There is therefore an increased risk of recurrence or progression of symptoms if there is delay in the diagnosis and initiation of preventative treatment. Similar trends appear to parallel in rates of admissions of other serious conditions including cardiac and surgical emergencies [19–22].

Although, there are multiple reports documenting the decrease in stroke admission rates during the pandemic, there is little research on stroke rates once the number of COVID-19 cases begins to decrease. A study of disease trends during the months following controlling the pandemic, as the cases of COVID-19 begin to decrease in the community, may offer important information on the reasons for the decline in stroke admissions during the pandemic. We present the state-wide admission rates for stroke from the Qatar Stroke Database for the entire year and compare stroke admissions to COVID-19 cases during the same time.

## Methods

The Qatar Stroke Database prospectively collects information on all acute stroke patients admitted to the Hamad General Hospital (HGH) in Qatar. The hospital is a tertiary stroke center that has a dedicated stroke ward. The hospital is the only center where stroke-specific treatments, including intravenous thrombolysis and mechanical thrombectomy are offered and where 98% of stroke patients requiring admission in Qatar are managed. The details of the database have previously been published [22, 23].

### Ethical approval and Patient Consent

No patient consent was required due to the study design.

The study was approved by the Institutional Review Board, Hamad Medical Corporation at the Medical Research Centre (MRC-16016/16).

All stroke patients admitted to HGH stroke service from September 2019 to December-2020 were evaluated for this study. In addition to overall rates of admissions, we evaluated the frequency of strokes subtypes, TIAs and stroke mimics during the study period. We reviewed presumed stroke admissions in three time periods; Sept 2019 to Feb 2020 (pre-pandemic), March to June 2020 (Pandemic) and July to December 2020 (post-peak of pandemic) 2020.

Clinical information including rates of admissions, clinical presentation, NIHSS at admission and discharge, TOAST classification, ethnicity, risk factors, investigations, complications,

length of stay and discharge outcomes were obtained. The modified Rankin Scale (mRS) pre-admission, at discharge, and at 90-day follow-up was also collected when available.

Descriptive and inferential statistics were used to characterize the study sample. Descriptive results (including graphical displays) for all quantitative variables (e.g., age) are presented as the mean ± standard deviation (SD) for normally distributed data or median with interquartile range for data not normally distributed. The monthly rate of stroke admissions was compared in the six months pre-Covid-19 (Jan-Feb 2020), four months when the pandemic cases were very high (March-June 2020) and for the following 6 months (July-Dec 2022). Bivariate analysis comparing the demographic and clinical characteristics of the patients along with some discharge outcomes during the study periods were performed using Independent sample t-test, and the Mann Whitney U-test to compare the average for all quantitative variables (e.g., age) between stroke subtypes, wherever appropriate, while the Pearson Chi-Square test or Fisher Exact test were used, as appropriate, in comparing all the qualitative variables. The statistical tests were performed using IBM SPSS Statistics ver. 27 (IBM, Armonk, USA). A p-value of 0.05 or less was considered significant.

## Results

There were 3162 stroke admissions to the HGH between September 2019 and December 2020 [age: 53.3 ±14.5; male: 2306 (72.9%) (Table 1). The younger age and higher percentage of males reflects the demographics of Qatar with a predominantly young male expatriate population. The monthly rates of COVID-19 diagnosis for the state of Qatar during the pandemic are shown in Fig 1 with the peak rates during March-June months. The monthly rates of stroke admissions during the year are shown in Table 1. There was a highly significant decrease in stroke admissions during the peak months of the pandemic (Table 1 and Fig 1). The significant decrease in the number of patients with atrial fibrillation is likely due to fewer patients being offered Holter monitors during the pandemic. The young age of the patients may also account for fewer diagnosis of atrial fibrillation as we have previously reported [23]. We have no clear explanation for the significantly fewer patients with diabetes or patients who were 'current smokers'. Perhaps the pandemic may have been a factor in fewer smokers in the pandemic and post-peak time period. The low mortality is likely related to the young age of the patient population and the very high incidence of lacunar stroke with mild symptoms as has been previously reported [21, 22]. The majority of strokes were seen in non-Qatari nationals (less than 20% patients were Qataris.

The decrease in the rates of medical and functional mimics evident during the peak months of the pandemic gradually returned towards pre-pandemic levels as the rates of COVID-19 cases decreased but never quite reached the baseline.

Using the TOAST criteria [24] for classifying the etiology, we noticed a significant decrease in the percentage of SVD in patients with confirmed ischemic stroke, decreasing from 44.1% in the pre-pandemic to 34.3% during the pandemic and 39.4% in the last six months of observation. We also noted a concomitant significant increase in the percentage of stroke due to large vessel disease, increasing from 14.4% in the pre-COVID 6 months to 24.8% in the 3 months in March-June and 21.3% in the post-peak months (see Table 1 and S1 Table). The details of the baseline, study period, diagnosis, TOAST classification, prognosis and interventions in two major ethnic groups of the study population, for the pre-COVID-19, during the peak and in the six months following the peak of the pandemic are provided in S1 Table. There were very few patients from Europe/USA, Africa and the Far East and therefore were not included in the analysis. We have previously reported on the significant differences in age, type of strokes and prognosis between Arab and South Asian patients [24, 25].

**Table 1. The details of the admissions, risk factors, TOAST classification, prognosis and interventions is shown for the pre-COVID-19, during the peak of the pandemic and in the six months following the peak of the pandemic.**

| Characteristic or Investigation | Total Patients | Pre-COVID Time (Sep 2019-Feb 2020) | Peak COVID Time (Mar-Jun 2020) | Post-Peak COVID Time (July-Dec 2020) | P- Value |
|---|---|---|---|---|---|
| | (n = 3162) | (n = 1376, average = 229.3/ month) | (n = 630, average = 157.5/ month) | (n = 1156, average = 192.7/month | |
| Age- Mean, years | 53.3 ±14.5 | 52.5 ±14.0 | 54.09 ±14.6 | 53.9 ±14.9 | 0.02 |
| Sex—Males | 2306 (72.9) | 1010 (73.4) | 468 (74.3) | 828 (71.6) | 0.42 |
| Diabetes | 1536 (48.6) | 626 (45.5) | 321 (51.0) | 586 (51.0) | 0.01 |
| Hypertension | 2040 (64.5) | 904 (65.7) | 409 (64.9) | 727 (62.9) | 0.33 |
| Prior Stroke | 368 (11.6) | 153 (11.1) | 68 (10.8) | 147 (2.7) | 0.35 |
| Coronary Artery Disease | 336 (10.6) | 142 (10.3) | 77 (12.2) | 117 (10.1) | 0.34 |
| Prior Atrial Fibrillation | 292 (9.2) | 99 (7.2) | 46 (7.3) | 147 (12.7) | <0.001 |
| Current Smoker | 677 (21.4) | 331 (24.1) | 118 (18.7) | 228 (19.7) | 0.006 |
| **Final Diagnosis** | | | | | |
| Ischemic Stroke | 1433 (45.3) | 572 (41.6) | 330 (52.4) | 531 (45.9) | <0.001 |
| Transient Ischemic Attack | 299 (9.5) | 157 (11.4) | 56 (8.9) | 86 (7.4) | |
| Intracerebral Hemorrhage | 276 (8.7) | 111 (8.1) | 69 (11.0) | 96 (8.3) | |
| Cerebral Venous Sinus Thrombosis | 52 (1.6) | 13 (0.9) | 15 (2.4) | 24 (2.1) | |
| Stroke Mimic | 1102 (34.9) | 523 (38.0) | 160 (25.4) | 419 (36.2) | |
| **TOAST Classification** | | | | | |
| Small Vessel Disease | 582 (40.1) | 258 (44.1) | 115 (34.3) | 209 (39.4) | 0.005 |
| Large Vessel Disease | 279 (19.2) | 84 (14.4) | 82 (24.5) | 113 (21.3) | |
| Cardio-Embolic | 388 (26.7) | 151 (25.8) | 97 (29.0) | 140 (26.4) | |
| Stroke of Determined Origin | 72 (5.0) | 32 (5.5) | 13 (3.9) | 27 (5.1) | |
| Stroke of Undetermined Origin | 130 (9.0) | 60 (10.3) | 28 (8.4) | 42 (7.9) | |
| **Prognosis–At Discharge** | | | | | |
| Good (mRS 0–2) | 2134 (67.5) | 997 (72.5) | 368 (58.4) | 769 (66.5) | <0.001 |
| Poor (mRS 3–6) | 1028 (32.5) | 379 (27.5) | 262 (41.6) | 387 (33.5) | |
| Mortality–At Discharge | 38 (1.2) | 15 (1.1) | 11 (1.7) | 12 (1.0) | 0.37 |
| **NIHSS** | | | | | |
| NIHSS on admission (mean) | 4.1 ±6.3 | 3.6 ±5.8 | 5.3 ±7.2 | 4.1 ±6.3 | <0.001 |
| Mild Stroke (NIHSS 4 or less) | 2344 (74.1) | 1071 (77.8) | 422 (67.0) | 851 (73.6) | <0.001 |
| Moderate Stroke (NIHSS 5–10) | 415 (13.1) | 155 (11.3) | 98 (15.6) | 162 (14.0) | |
| Severe Stroke (NIHSS > 10) | 403 (12.7) | 150 (10.9) | 110 (17.5) | 143 (12.4) | |
| **Intervention** | | | | | |
| IV Thrombolysis | 138 (4.4) | 60 (4.4) | 34 (5.4) | 44 (3.8) | 0.29 |
| Door to Needle Time (minutes) | 68.4 ±31.0 | 67.7 ±34.2 | 68.8 ±32.7 | 68.9 ±32.7 | 0.97 |
| Interventional Thrombectomy | 40 (1.3) | 20 (1.5) | 10 (1.6) | 10 (0.9) | 0.30 |

The severity of symptoms at presentation increased during the pandemic months when compared to the pre and post-peak pandemic time period. There was an increase in the average NIHSS on admission from 3.6 ±5.8 in the pre-COVID 6 months to 5.3 ±7.2 in March-June and decreasing to 4.1 ±6.3in the following 6 months. The percentage of patients with mild symptoms (NIHSS ≤ 4) decreased from 1071 (77.8%) in the pre-COVID months to 422 (67.0%) in Feb-June and increased to 851 (73.6%) in the last six months of 2020 (p = 0.001).

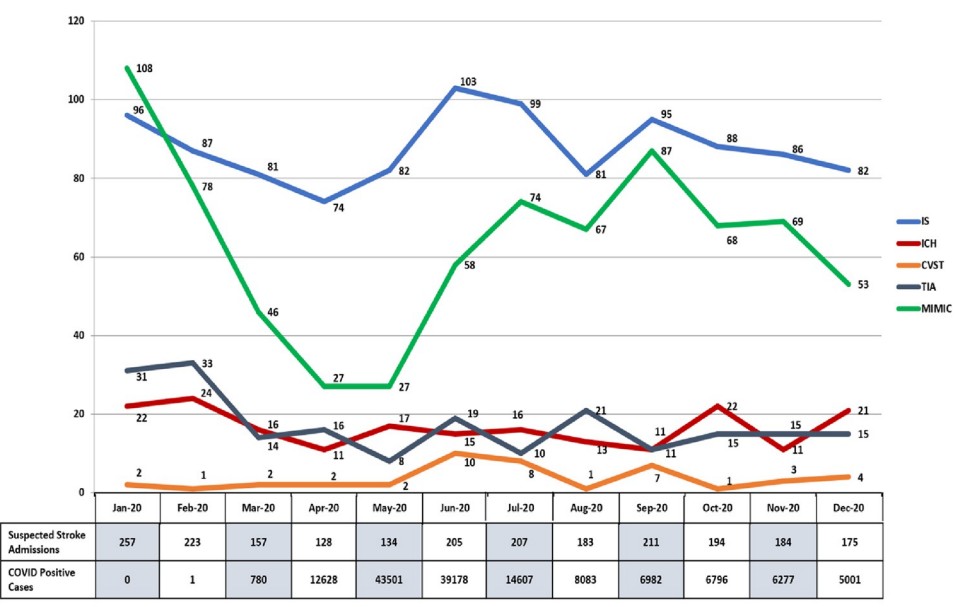

IS: Ischemic Stroke

ICH: Intracerebral hemorrhage

CVST: Cerebral venous sinus thrombosis

TIA: transient ischemic attack

Mimics

**Fig 1. Distribution of different discharge diagnosis per month during the pre-COVID-19, peak of the pandemic and in the six months following the peak of the COVID-19 pandemic.** The number of monthly COVID-19 cases are shown in the legend below. The "P" values are for group comparisons between the pre-pandemic, pandemic and post-pandemic time period.

There was also a significant increase in the rates of patients with poor functional recovery during the four months (March-June 2020) when compared to the pre and post pandemic time. The number of patients with mRS of 0–2 at discharge decreased from 997 (72.5%) in the pre-COVID months to 368 (58.4%) in the four months post-COVID and improved again in the following 6 months to 769 (66.5%) with p = 0<0.001.

## Discussion

Our study supports previous reports documenting the dramatic decline in acute stroke admissions during the COVID-19 pandemic. We also provide evidence that while there was a return towards increasing admissions as the pandemic conditions improved, it did not return to the pre-pandemic levels. Our previous report outlined the dramatic decrease in the stroke rates during the three months of the pandemic, driven predominantly by reduction in stroke mimics. The admissions for ischemic stroke and ICH had remained unchanged during the height of the pandemic [14]. This led to a relative increase in the admission of patients with severe symptoms leading to poorer functional recovery [14]. Following the height of the pandemic, as the number of COVID-19 cases began to fall, the number of suspected stroke admissions to the hospital began to increase. Once again, this was predominantly driven by the increasing number of stroke mimic admissions. Interestingly, although the number of COVID-19 cases

decreased significantly in July-December 2020, the numbers remained higher than 197/ month. Similarly, although stroke admissions increased, these did not reach the pre-pandemic levels.

The effects of lockdown in the recent COVID-19 pandemic have been reported on several acute emergencies [12, 19–21, 26]. In most regions the decrease in stroke and acute coronary syndromes (ACS) have been similar although in Greece, there was a decrease in acute stroke but not ACS [27]. In Finland, there was a reduction in the rates of several acute medical illnesses seen in the emergency department, including infections (28%), back or limb pain (31%) and psychiatric illness (19%). Interestingly, there was no decrease in the number of stroke or ACS admissions during the period of observation [21]. In Qatar, decline in admissions to the emergency department varied between 9% to 75% was observed for acute surgical emergencies, ACS, bone fractures and cancer whereas admissions for respiratory conditions increased [20].

Several studies noted that the severity of symptoms of stroke patients increased during the pandemic [28]. There are reports of formation of recurrent thrombi during treatment of acute stroke [28]. An enhanced thrombotic state has been reported with COVID-19 infection and this may explain the higher incidence of severe strokes [29–31]. In addition, the COVID-19 virus may directly damage the cerebral vascular endothelium, making it more thrombogenic [30, 32]. The increased severity of stroke symptoms may also be artificially increased because of the frequent observation that the majority of patients with TIAs and mild stroke are not coming to hospital as is evident from our data and the reported decline in milder cases in published reports [4,8, 11, 13, 14].

To our knowledge, there is only one other study that reported on the stroke rates following the pandemic [33]. Similar to our observations, in the report from Norway, stroke admissions following the height of the pandemic did return to the pre-pandemic levels once the COVID-19 pandemic was under better control. In Norway, the decrease was evidence for TIAs and mild stroke but not for ICH during the lockdown period [33]. Similar to our report, the severity of symptoms was higher during the lockdown and there were delays in admission to hospital [33]. The reasons for the decreased admissions following the peak COVID-19 may be persistent avoidance behavior and concerns over contracting COVID-19 even when the numbers are down or a true decrease in the stroke rates in the community. Factors that may contribute to the lower rates include social distancing, reduced other infections, decreased air pollution or other unmeasured measures that require further study.

The dramatic decline in rates of strokes and other illnesses to hospital requires further study and has been a subject of much speculation. The most prevalent hypotheses relate to a fear for coming to hospital for fear of infection. This in turn may be magnified from 'stay-at-home' orders, leading to deferring urgent care as a recent survey from UK [34]. In Germany, the initial early decline in stroke-related consultations in the pandemic and later increase for telemedicine paralleled the population activities during lockdowns [35]. Another study from France also reported that there appeared to be a relationship between the decrease in stroke admissions to the severity of the COVID-19 illness [36].

Several additional factors proposed to explain the decrease in emergency visits of acute illnesses and the lower rates even when the restrictions were lifted as the cases decreased. A decrease in physical activity during lockdown may also have potential protective effects. An increase in physical activity is known to increase blood pressure potentially increasing the risk of SAH [37] and ACS [38]. Several virus and bacterial infections are known to trigger plaque rupture acute heart failure, ACS and acute stroke [39]. An intriguing hypothesis suggests that the diminished cardiovascular disease during the COVID-19 pandemic may be secondary to the extreme reduction in non-COVID-19 related community-acquired infections [40]. There

has been an almost 50% reduction in the diagnosis of new-onset atrial fibrillation in Denmark following the lockdown and as atrial fibrillation is an important stroke mechanism in the elderly, may also potentially contribute to the low rates of strokes [41].

In summary, the evidence from the Qatar Database shows that there is a gradual return of the rates of stroke admissions to the hospital as the COVID-19 cases began to decrease in the community. This was predominantly driven by an increase in stroke mimics. Interestingly, although the COVID-19 infections fell significantly in the months following June, 2020, the rates of stroke admissions did not return to the pre-pandemic levels. The reasons for the for the decreasing stroke numbers is unclear and the possible mechanism are discussed above.

## Supporting information

**S1 Table. Baseline characteristics of study population.**
(DOCX)

## Acknowledgments

We acknowledge the assistance of all involved physicians, nurses, and staff of the Stroke Team in HMC. We also thank Ms. Reny Francis (HMC) and Kath McKenzie (University of Alberta) for her editorial assistance and supportive care.

## Author Contributions

**Conceptualization:** Naveed Akhtar, Saadat Kamran, Ashfaq Shuaib.

**Data curation:** Yahia Imam, Sujatha Joseph, Deborah Morgan, Mohamed Abokersh, R. T. Uy.

**Formal analysis:** Naveed Akhtar.

**Supervision:** Ashfaq Shuaib.

**Validation:** Yahia Imam, Ashfaq Shuaib.

**Writing – original draft:** Naveed Akhtar, Ashfaq Shuaib.

**Writing – review & editing:** Naveed Akhtar, Saadat Kamran, Salman Al-Jerdi, Yahia Imam, Sujatha Joseph, Deborah Morgan, Ashfaq Shuaib.

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
