## [Decision Letter · Decision Letter 0]

29 Jun 2021

PONE-D-21-09805

Trends in stroke admissions before, during and post-peak of the COVID-19 pandemic: A one-year experience from the Qatar stroke database.

PLOS ONE

Dear Dr. Shuaib,

Thank you for submitting your manuscript to PLOS ONE. After careful consideration, we feel that it has merit but does not fully meet PLOS ONE’s publication criteria as it currently stands. Therefore, we invite you to submit a revised version of the manuscript that addresses the points raised during the review process.

We look forward to receiving your revised manuscript.

Kind regards,

Chiara Lazzeri

Academic Editor

PLOS ONE

Journal Requirements:

2. Thank you for submitting the above manuscript to PLOS ONE. During our internal evaluation of the manuscript, we found significant text overlap between your submission and the following previously published works, some of which you are an author.

- https://neurologyopen.bmj.com/content/3/1/e000084

The text that needs to be addressed involves the Results Section.

Please revise the manuscript to rephrase the duplicated text, cite your sources, and/or provide details as to how the current manuscript advances on previous work. Please note that further consideration is dependent on the submission of a manuscript that addresses these concerns about the overlap in text with published work.

We will carefully review your manuscript upon resubmission, so please ensure that your revision is thorough."

3. Thank you for including your ethics statement:  "Hamad Medical Corporation IRB approved the study 16016/16, a quality improvement study . This article is based on the data from this study".   

Please provide additional details regarding participant consent. In the ethics statement in the Methods and online submission information, please ensure that you have specified (1) whether consent was informed and (2) what type you obtained (for instance, written or verbal, and if verbal, how it was documented and witnessed). If your study included minors, state whether you obtained consent from parents or guardians. If the need for consent was waived by the ethics committee, please include this information.

Additional Editor Comments (if provided):

Reviewers' comments:

Reviewer's Responses to Questions

**Comments to the Author**

1. Is the manuscript technically sound, and do the data support the conclusions?

Reviewer #1: Yes

Reviewer #2: Yes

2. Has the statistical analysis been performed appropriately and rigorously? 

Reviewer #1: Yes

Reviewer #2: I Don't Know

3. Have the authors made all data underlying the findings in their manuscript fully available?

Reviewer #1: Yes

Reviewer #2: Yes

4. Is the manuscript presented in an intelligible fashion and written in standard English?

Reviewer #1: Yes

Reviewer #2: Yes

5. Review Comments to the Author

Reviewer #1: Important and straightforward observation of decline and rise of stroke admissions during and after the pandemic.

Some minor points to consider:

-was there an influence of vaccinations on the rise of cases?

-it is stated that 98% of all cases are admitted. This seems to be the number for all hospitalized cases? Please clarify.

-the overall young age and male preponderance needs special mentioning although the relative rates did not change over time. Also, why so few females seek hospital admission.

-no mention of minorities or non-Qataris is a clear weakness of this study and should be addressed. Where more severe strokes as seen during the pandemic due to higher admission rates of non-Qataris?

-the low mortality of around 1% needs explanation.

-Table 1 should denote that p values are for group comparisons between before and after and not other groups in the table. Or was it p for trend?

Reviewer #2: This is an interesting article examining trends in stroke admission 6 months before, 4 months during and 6 months after the COVID -19 pandemic. One question - the authors state that the numbers of Afib patients was likely fewer due to fewer holter monitors being placed during the pandemic, however can you expand this thought? How many stroke patients are routinely diagnosed as having afib as an outpatient with a Holter Monitor? Presumably most patients with Afib would still be diagnosed from their telemetry monitoring.

6. PLOS authors have the option to publish the peer review history of their article (what does this mean?). If published, this will include your full peer review and any attached files.

Reviewer #1: **Yes: **Michael Brainin MD

Reviewer #2: No

---

## [Author Response · Author response to Decision Letter 0]

8 Jul 2021

Response to Reviewers

Some minor points to consider:

1. was there an influence of vaccinations on the rise of cases? 

This is a good question but unfortunately we were unable to evaluate the effect of vaccinations on the rates of stroke in our study.

2. it is stated that 98% of all cases are admitted. This seems to be the number for all hospitalized cases? Please clarify.

The Hamad General Hospital (HGH is a tertiary care facility in the State of Qatar. The Stroke Program with the thrombolysis and thrombectomy facilities for the Stroke of Qatar are located at the HGH. Almost all acute strokes (98%) in Qatar are transferred to the hospital for treatment. We have made changes in the manuscript to make it clear for the readers.

3. the overall young age and male preponderance needs special mentioning although the relative rates did not change over time. Also, why so few females seek hospital admission.

We have made changes in the manuscript to explain that more than 70% of the population in Qatar comprises of young age (less than 55 years), with 5:1 expatriate males/female ratio. The younger age and higher percentage of male patients reflects these demographics. 

4. no mention of minorities or non-Qataris is a clear weakness of this study and should be addressed. Where more severe strokes as seen during the pandemic due to higher admission rates of non-Qataris?

We thank the reviewer for the comment. We have added a supplemental table to explain the “minorities and non-Qatari” demographics

5. the low mortality of around 1% needs explanation.

The low mortality is secondary to predominantly mild sub-cortical stroke in younger population. We have explained this in the results section

6. Table 1 should denote that p values are for group comparisons between before and after and not other groups in the table. Or was it p for trend?

We have made the correction in the legend of the table.

Reviewer #2: This is an interesting article examining trends in stroke admission 6 months before, 4 months during and 6 months after the COVID -19 pandemic. 

One question - the authors state that the numbers of Afib patients was likely fewer due to fewer holter monitors being placed during the pandemic, however can you expand this thought? How many stroke patients are routinely diagnosed as having afib as an outpatient with a Holter Monitor? Presumably most patients with Afib would still be diagnosed from their telemetry monitoring.

We thank the reviewer for the question. We have made changes to the results section to explain that the lower atrial fibrillation rates are likely because of the younger age of the patients. We have previously reported this and have included the reference in the manuscript.

---

## [Editor Report · Decision Letter 1]

12 Jul 2021

Trends in stroke admissions before, during and post-peak of the COVID-19 pandemic: A one-year experience from the Qatar stroke database.

PONE-D-21-09805R1

Dear Dr. Shuaib,

We’re pleased to inform you that your manuscript has been judged scientifically suitable for publication and will be formally accepted for publication once it meets all outstanding technical requirements.

Kind regards,

Chiara Lazzeri

Academic Editor

PLOS ONE
---

## [Editor Report · Acceptance letter]

10 Mar 2022

PONE-D-21-09805R1 

Trends in stroke admissions before, during and post-peak of the COVID-19 pandemic: A one-year experience from the Qatar stroke database. 

Dear Dr. Shuaib:

I'm pleased to inform you that your manuscript has been deemed suitable for publication in PLOS ONE. Congratulations! Your manuscript is now with our production department. 

Kind regards, 

on behalf of

Dr. Chiara Lazzeri 

Academic Editor

PLOS ONE